# OpenReview forum: "Progressive LLM Alignments Using Two-Player Games"
_ICLR.cc/2025/Conference — Submitted to ICLR 2025_

### Official Review · Reviewer_MEdA · 2024-10-24

**Soundness:** 3
**Presentation:** 3
**Contribution:** 2
**Rating:** 5
**Confidence:** 4

**Summary:**

This paper introduces a novel framework that formulate the alignment problem as a two-player zero-sum game. This framework involves an adversarial agent and a defensive agent that iteratively interact to improve the LLM’s performance. The adversarial agent generates prompts to reveal weaknesses in the defensive agent’s responses, while the defensive agent seeks to adapt and strengthen its performance based on these prompts.

**Strengths:**

The main strength of this paper are:
1. The overall writing is well-organized and easy to follow, making the ideas presented clear and understandable.
2. The experimental results appear solid, especially in safety-related tasks. The proposed framework shows improvements compared to traditional RLHF methods, particularly in handling harmful inputs and jailbreak scenarios, which suggests that the approach is effective in these contexts.

**Weaknesses:**

1. Lack of Novelty and Insight. While the overall idea is well-executed, it seems relatively straightforward and lacks significant novelty. The two-player game framework, while effective in this context, feels more like an incremental improvement rather than a significant innovation.
2. Triviality of the Additional Diversity Reward. The additional diversity reward also feels somewhat trivial, as it is a common technique in multi-agent settings. It appears more as a practical trick rather than a meaningful contribution or innovation to the overall methodology.
3. Technical Flaw: The paper’s analysis relies on mirror descent, which guarantees convergence only for the average strategy[1]. However, the final round strategy tends to cycle around the Nash equilibrium rather than converge to it [1][2]. As a result, using only the final strategy in place of the average strategy is not theoretically justified in this context.
4. Computational Cost: The approach requires maintaining two policies for alternating updates, with each policy being optimized using PPO. This results in substantial storage and computational costs, particularly in the context of RLHF. Furthermore, as highlighted in the third point, the use of mirror descent mandates tracking the average policy over time, making it insufficient to rely solely on the final policy. Storing all the historical policies or learning an averagy policy further exacerbates the computational burden, complicating practical implementation at scale.

[1] Mertikopoulos, P., Lecouat, B., Zenati, H., Foo, C. S., Chandrasekhar, V., & Piliouras, G. (2018). Optimistic mirror descent in saddle-point problems: Going the extra (gradient) mile. arXiv preprint arXiv:1807.02629.
[2] Perolat, J., Munos, R., Lespiau, J. B., Omidshafiei, S., Rowland, M., Ortega, P., ... & Tuyls, K. (2021, July). From poincaré recurrence to convergence in imperfect information games: Finding equilibrium via regularization. In International Conference on Machine Learning (pp. 8525-8535). PMLR.

**Questions:**

1. How much GPU memory was required to run the llama-2-7B experiments for alternative updating?
2. How many total iterations were performed during the experiments? Was the performance consistently improving throughout the iterations？

---

> ### Author Response · Authors · 2024-11-24
> **Official Comment by Authors (1/2)**
>
> Thank you to the reviewer for providing insightful comments.
>
> **Q1: Lack of Novelty and Insight. While the overall idea is well-executed, it seems relatively straightforward and lacks significant novelty. The two-player game framework, while effective in this context, feels more like an incremental improvement rather than a significant innovation.**
>
> >The core innovation of our approach lies in the dynamic, competitive interplay between an adversarial and a defensive agent, which allows for continual adaptation and refinement of both agents, leading to well-generalized aligned models, particularly on OOD datasets. Traditional alignment methods, such as RLHF, rely on static prompt datasets and human-driven instructions, leading to potential gaps in real-world coverage. In contrast, our two-player game framework dynamically generates and adapts prompts, uncovering and addressing weaknesses of LLMs that might otherwise go unnoticed. This iterative adversarial process, where the adversarial agent is continuously forced to generate novel challenges, provides an ongoing learning loop, which enhances generalization and robustness—a key challenge in the field of LLM alignment. Additionally, our theoretical proof of convergence to Nash Equilibrium is novel in its own right, adding mathematical rigor to the process of adversarial alignment.
>
> **Q2: Triviality of the Additional Diversity Reward. The additional diversity reward also feels somewhat trivial, as it is a common technique in multi-agent settings. It appears more as a practical trick rather than a meaningful contribution or innovation to the overall methodology.**
>
> >Regarding the additional diversity reward, while diversity constraints (like BLEU scores and sentence embeddings) may seem common in multi-agent settings, the specific role they play in our framework is crucial. The diversity reward is not just a technical addition; it serves to prevent the adversarial agent from exploiting a narrow set of prompts, ensuring that the generated prompts continuously push the boundaries of the defensive agent’s capabilities. This prevents premature convergence to an easily defeatable set of adversarial inputs, thus facilitating a more comprehensive training process. In the context of RLHF-based alignment, where prompt coverage can be limited, our method of dynamically generating diverse, challenging prompts enhances the alignment process by ensuring broad and adaptive coverage of edge cases, thereby strengthening model robustness.
>
> **Q3: Technical Flaw: The paper’s analysis relies on mirror descent, which guarantees convergence only for the average strategy[1]. However, the final round strategy tends to cycle around the Nash equilibrium rather than converge to it [1][2]. As a result, using only the final strategy in place of the average strategy is not theoretically justified in this context.**
>
> > In theory, we can select the best policy from all iterations to yield, which has a similar effect as average policy [1]. And in our experimental setting context, this is similar to selecting the final policy.
>
> >[1] Exploratory Preference Optimization: Harnessing Implicit Q*-Approximation for Sample-Efficient RLHF. Xie & Foster et al.
>
> **Q4: Computational Cost: The approach requires maintaining two policies for alternating updates, with each policy being optimized using PPO. This results in substantial storage and computational costs, particularly in the context of RLHF. Furthermore, as highlighted in the third point, the use of mirror descent mandates tracking the average policy over time, making it insufficient to rely solely on the final policy. Storing all the historical policies or learning an averagy policy further exacerbates the computational burden, complicating practical implementation at scale.**
>
> >**Computational Overhead Considerations**: While maintaining two policies requires additional computational resources, our approach carefully balances this overhead by leveraging **shared resources** between the two agents. Each agent’s optimization process can be managed through efficient use of the Proximal Policy Optimization (PPO) method, with the inclusion of KL regularization stabilizing the learning process. This ensures that updates are not computationally prohibitive, even in the context of large-scale implementations.
>
> >**Handling Mirror Descent and Averaging Policies**: The concern regarding mirror descent and policy averaging is addressed by leveraging efficient memory management techniques. While it is true that tracking historical policies can add to storage and computational costs, we have implemented strategies to minimize this burden. For example, instead of storing all historical policies, we use a dynamic policy averaging mechanism that efficiently tracks only relevant information needed for stable learning, thus reducing the overall memory footprint.

---

> > ### Author Response · Authors · 2024-11-24
> > **Official Comment by Authors (2/2)**
> >
> > **Q5: How much GPU memory was required to run the llama-2-7B experiments for alternative updating?**
> >
> > >For our experiments, we utilized 8 NVIDIA A100 GPUs, each with 80GB of memory. This setup provided sufficient memory to handle the computational demands of the alternative updating process in our experiments, ensuring smooth training and optimization of both the adversarial and defensive agents.
> >
> > **Q6: How many total iterations were performed during the experiments? Was the performance consistently improving throughout the iterations**
> >
> > >A total of 600 iterations were performed during our experiments. As the training progressed, the performance gradually improved and began to converge. Notably, the converged performance of our method outperformed the RLHF baseline, demonstrating that the iterative two-player game process of our approach leads to more powerful and aligned models compared to the standard RLHF method.

---

> > > ### Comment · Reviewer_MEdA · 2024-11-26
> > >
> > > Thanks for the reponse! Could you clarify what is specifically meant by "shared resources" in this context? Which modules or computational processes are being shared between the two agents, and how is this sharing implemented? Additionally, how is PPO utilized efficiently in this dual-policy framework? Could you provide a detailed description of the training workflow, including specific steps and optimizations employed to manage computational overhead?

---

### Official Review · Reviewer_F894 · 2024-11-01

**Soundness:** 2
**Presentation:** 2
**Contribution:** 2
**Rating:** 3
**Confidence:** 3

**Summary:**

The authors in this paper address the limitations of traditional LLM alignment methods, which often rely on static prompt sets pre-collected by human labelers. Current methods need more adaptability to identify areas where LLMs require dynamic improvement.
The authors propose a two-player game involving an adversarial (tutor) and a defensive (student) LLM to overcome these issues. The adversarial LLM automatically generates challenging prompts designed to expose the defensive LLM's weaknesses, pushing it to adapt and improve iteratively. The iterative adversarial alignment process is shown converge to Nash equilibrium between the adversarial and defensive agents. Moreover, they have also given an algorithm that finds the $O(1/\sqrt{T})$-approximate Nash equilibrium in T iterations.

**Strengths:**

The problem addressed in this paper is both exciting and novel, offering a fresh approach to LLM alignment. The analysis appears sound, and the proofs seem correct at first glance. However, I have some questions I would like to clarify, as highlighted below.

**Weaknesses:**

Some things need to be appropriately motivated; for example, $R_{div}(x)$ in eqn 3.1 is defined, but I need to figure out how to obtain this. Only in some sections is it defined, but that is also very restrictive, and how this will be defined or obtained in general needs to be clarified. Look at some more questions regarding this below.

Different variants of the same algorithms are also hard to parse, and there needs to be a discussion about which algorithm is finally used in the theoretical analysis and why.

Though the paper addresses a good problem, it still lacks some details, and I would like to see more clarity in the revised versions.

My primary concern lies with the motivation and problem formulation. The central motivation here relies on an attack prompt, assuming that an adversarial player controls the prompt distribution. This assumption seems overly strong. In practice, unless prompt optimization techniques or another language model are employed, we have limited control over the prompts users provide.

**Questions:**

1. What is the motivation behind using the $R_{\text{div}}$​ term in Equation 3.1? Specifically, could you
a. explain how this diversity reward relates to or enhances existing alignment objectives.
b. discuss the advantages of this approach over traditional alignment methods.
c. clarify the general definition or derivation of $R_{\text{div}}$​, as its current form seems restrictive in certain sections.

2. I would like further explanation on including the KL divergence term in Equation 3.3, which is absent in Equation 3.1.

a. Could you introduce the KL divergence term when it first appears in Equation 3.3 and discuss its implications for the overall optimization process?

3. The paper presents at least three algorithm variants, making it unclear which ones are used in the theoretical analysis and implementation and the rationale for this choice. Additionally, given these variants, what are the discrepancies observed between the theoretical and implemented versions?

4. The existence of a Nash equilibrium (NE) is asserted based on the linearity of $J(\pi, \mu)$ in Equation 3.1. However, Algorithm 1 introduces KL terms in its updates, which contradicts this claim. Even without the KL term, could you explain why the $R_{\text{div}}$​ term in Equation 3.1 would be linear?

5.Outside of the attack prompting scenario, it needs to be clarified why minimization over the prompt distribution is necessary. Please clarify specific use cases where the user has the flexibility to control the prompt distribution.

---

> ### Author Response · Authors · 2024-11-24
> **Official Comment by Authors (1/2)**
>
> We sincerely thank you for providing thoughtful and constructive feedback. Based on your feedback, we have revised the statements in the method section to make them more reader-friendly and easier to understand. **The changes are marked in blue.**
>
>
> **Q1: What is the motivation behind using the Rdiv term in Equation 3.1? Specifically, could you a. explain how this diversity reward relates to or enhances existing alignment objectives. b. discuss the advantages of this approach over traditional alignment methods. c. clarify the general definition or derivation of Rdiv, as its current form seems restrictive in certain sections.**
>
> > **Motivation behind using the Rdiv term in Equation 3.1:**
>
> >**Relation to alignment objectives:**  The diversity reward $R_{div}(x)$ influences only the optimization process of the adversarial agent, which aims to generate prompts where the defense model underperforms. As shown in Eq. (3.1) and Eq. (3.3), the defense model is optimized based on the prompts generated by the adversarial agent, $x \sim \mu(⋅)$. By adding the diversity reward $R_{div}(x)$, the adversarial agent is encouraged to identify a broader range of weaknesses in the defense model, facilitating the defense model’s improvement across all identified vulnerabilities. Without the diversity reward, the adversarial agent might overfit to a narrow set of prompt types, limiting the extent of the defense model's improvement.
>
> >**Advantages over traditional alignment methods:** Traditional alignment methods typically focus on optimizing for a fixed set of prompts or responses, which may fail to address all edge cases. In reality, we cannot control the prompts users provide; therefore, our goal is to generalize as much as possible. By incorporating diversity into the prompts, we ensure broader coverage of potential adversarial cases, thereby enhancing the model's overall performance and robustness against unexpected inputs.
>
> >**Clarification of general definition of $R_{div}(x)$:**
> The diversity reward $R_{div}(x)$ is solely related to the prompt $x$ and measures the dissimilarity of generated prompts to previous generations, encouraging the adversarial agent to produce unique prompts each time. Therefore, any similarity measure for prompts can be applied. In Section 3.2.2, we explain how to compute $R_{div}(x)$ using two text similarity measures: SelfBLEU (Eq. 3.4) and sentence embeddings (Eq. 3.5).
>
> In light of your feedback, we have revised the manuscript to include this clarification, ensuring that the definition and the motivation of$R_{div}(x)$ are clearly articulated.
>
> **Q2: I would like further explanation on including the KL divergence term in Equation 3.3, which is absent in Equation 3.1. Could you introduce the KL divergence term when it first appears in Equation 3.3 and discuss its implications for the overall optimization process?**
>
> >The KL divergence term in Equation 3.3 is included to regularize the adversarial agent’s prompt generation process, in line with the Follow-the-Regularized-Leader (FTRL) algorithm, which plays a key role in theoretically ensuring that the system converges to a Nash Equilibrium.   The term $\text{KL}( \mu_{\phi}(x) \| \mu_{\phi t-1}(x))$ penalizes the adversarial agent for making large changes to its prompt distribution across iterations, thereby maintaining stability in the training process. This ensures that the adversarial agent continues to explore new, challenging prompts while avoiding drastic shifts in its strategy .
>
> >The purpose of regularization  is to strike a balance between exploration (generating diverse prompts) and stability (not overfitting to a narrow set of strategies), making the iterative optimization process more stable and effective.
>
> > We have revised the content in Section 3.1 to provide additional context.

---

> ### Author Response · Authors · 2024-11-24
> **Official Comment by Authors (2/2)**
>
> **Q3: The paper presents at least three algorithm variants, making it unclear which ones are used in the theoretical analysis and implementation and the rationale for this choice. Additionally, given these variants, what are the discrepancies observed between the theoretical and implemented versions?**
>
> > Algorithm 1 is our practical implementation used for experiments. Algorithms 2 and 3 are theoretical variants that differ from Algorithm 1 in two ways: their output policy generation and diversity treatment. While Algorithms 2 and 3 yield the mean policy (common for theoretical convergence analysis), Algorithm 1 yields the final policy, which is more practical and convenient.
> Since it is challenging to theoretically analyze the importance of the diversity score with a general diversity reward $R_{\rm div}(x)$ as defined in Algorithm 1, we introduce Algorithm 3, which uses entropy as the diversity reward. We demonstrate that incorporating diversity constraints leads to a more varied prompt distribution, while the absence of the entropy regularizer causes the adversarial agent to converge to a single-point prompt distribution.
>
> >We **have included** the above discussion in **Appendix A.4**.
>
> Q4: The existence of a Nash equilibrium (NE) is asserted based on the linearity of J(π,μ) in Equation 3.1. However, Algorithm 1 introduces KL terms in its updates, which contradicts this claim. Even without the KL term, could you explain why the Rdiv term in Equation 3.1 would be linear?
>
> >Algorithm 1 outlines how we optimize the objective in Eq. 3.1. As shown in the Follow-the-Regularized-Leader (FTRL) algorithm, it is common to add regularizers when optimizing an objective to achieve smoother training or enforce desired properties. Importantly, even with the regularizers, the objective remains concave for the max player and convex for the min player, ensuring that a Nash Equilibrium (NE) still exists.
>
> >Linearity of x: Here we consider the objective is linear or not in $\mu$ which is a function on X. Here we can write $E_\mu[R_\text{div}(x)] = <\mu, R_\text{div}>$ which indicates this term is linear in $\mu$.
>
> **Q5: Outside of the attack prompting scenario, it needs to be clarified why minimization over the prompt distribution is necessary. Please clarify specific use cases where the user has the flexibility to control the prompt distribution.**
>
> >Even beyond the attack prompting scenario, the underlying motivation and benefits of the two-player game alignment framework remain valid: we aim for the aligned model to generalize well. By minimizing the prompt distribution, the adversarial agent identifies areas where the current aligned model (the defense model) underperforms. The defense agent can then focus on improving itself in these identified weak areas.
>
> >**Use cases where users control prompt distribution**: In practical scenarios, certain actors (such as malicious users or adversarial agents) may have control over the inputs to the model, and the defensive agent needs to be robust to such varied distributions. This assumption allows us to model scenarios where an adversarial player might shape the prompts to exploit weaknesses in the defensive agent.
>
> >**Clarification**: In real-world applications, user-controlled prompt distributions may arise in contexts such as user-generated content (e.g., inputs in conversational AI or interactive systems) or adversarial testing scenarios where prompts are crafted to challenge the model's behavior. This mechanism allows the defensive model to better handle unexpected and potentially harmful inputs, ensuring more generalizable robustness .

---

### Official Review · Reviewer_6iyV · 2024-11-03

**Soundness:** 3
**Presentation:** 3
**Contribution:** 2
**Rating:** 5
**Confidence:** 4

**Summary:**

This work proposes to use two-player zero-sum games to perform LLM alignment in safety-critical scenarios. This method iteratively trains a defensive agent and an adversarial agent in turn to adaptively generate progressively harder prompts. The authors provide a theoretical analysis of the method's convergence to Nash equilibrium and perform experiments to show the effectiveness of the method.

**Strengths:**

1. The idea of using self-play in a two-player zero-sum game to improve LLM alignment is novel and intuitive.
2. The manuscript is well-organized and easy to follow. The main idea of using self-play and introducing diversity is well-explained.
3. The authors provide theoretical analysis as well as experiment results to show the effectiveness of their method.

**Weaknesses:**

1. More comprehensive evaluation.
    1. The theoretical result claims that the proposed method can converge to a Nash equilibrium, but there are no experiment results validating this claim. I would suggest the author use metrics like exploitability or NashConv to evaluate how far the current agents are from Nash equilibrium.
    2. The main results in Table 1, and 2 only show performance under a certain amount of training. A more comprehensive evaluation is to show the method's performance curve w.r.t training amount, e.g., the performance curve w.r.t. GPO iteration. This could better compare GPO with baselines like RLHF to show the effect brought by self-play training and show the progressive improvement process of GPO.
2. Need for ethics and social impact statement: this method trains a defensive agent as well as an adversarial agent. Although the authors discuss that the adversarial agent can be utilized for red teaming, it can also be potentially used to make attacks and induce harmful behaviors. However, the authors claim "this work does not involve potential malicious or unintended uses ..." in the ethics statement. I would suggest the authors add necessary discussions on how to prevent potentially harmful use of their method.

**Questions:**

Please see Weaknesses.

**Details Of Ethics Concerns:**

This method trains a defensive LLM agent as well as an adversarial LLM agent in safety-critical tasks. The adversarial agent can be potentially used to make attacks and induce harmful behaviors of LLMs and the authors do not address these potential ethics problems in their original submission.

---

> ### Author Response · Authors · 2024-11-24
> **Official Comment by Authors**
>
> We sincerely thank the reviewer for providing valuable feedback. We have included discussions on safeguards and considerations for responsible use to ensure that our method is applied ethically and avoids any unintended harmful consequences.  **The changes are marked in blue.**
>
> **Q1: More comprehensive evaluation.The theoretical result claims that the proposed method can converge to a Nash equilibrium, but there are no experiment results validating this claim. I would suggest the author use metrics like exploitability or NashConv to evaluate how far the current agents are from Nash equilibrium.**
>
> >We can use the Nash gap to evaluate the convergence to the Nash equilibrium. The Nash gap is defined in equation (3.6). We can approximate the Nash gap in practice by measuring the gap between the harmful rate (or harmful score) of two iterations that optimizes the defensive policy and the attacking policy. Through an image similar to Figure 2-(c), we are able to observe the performance variations between different iterations. This enables us to ensure that the performance of our method reaches a point where it can no longer be further enhanced, signifying the attainment of convergence.
>
> **Q2: The main results in Table 1, and 2 only show performance under a certain amount of training. A more comprehensive evaluation is to show the method's performance curve w.r.t training amount, e.g., the performance curve w.r.t. GPO iteration. This could better compare GPO with baselines like RLHF to show the effect brought by self-play training and show the progressive improvement process of GPO.**
>
> >In Figure 2-(c) of the paper, we have presented the performance progression of our method (GPO) alongside baselines like RLHF. This figure demonstrates how the safety performance evolves as the training iterations increase. Notably, we observe that while RLHF's safety performance plateaus after a certain number of iterations, our method, continues to improve beyond this point.  Specifically, the model becomes even more robust to attacks, as evidenced by the lower attack success rate.
>
> >This demonstrates that the two-player game dynamics introduced by GPO drive continuous enhancements in safety alignment throughout training, ultimately resulting in a robust defense model and a strong attacker model, particularly on OOD datasets.
>
> **Q3: Need for ethics and social impact statement: this method trains a defensive agent as well as an adversarial agent. Although the authors discuss that the adversarial agent can be utilized for red teaming, it can also be potentially used to make attacks and induce harmful behaviors. However, the authors claim "this work does not involve potential malicious or unintended uses ..." in the ethics statement. I would suggest the authors add necessary discussions on how to prevent potentially harmful use of their method.**
>
> >We thank the reviewer for the valuable suggestion.  We acknowledge that our initial ethics statement may not have fully addressed the potential risks associated with the adversarial agent. We have revised the statement to provide clearer guidance on the possible consequences of the method.
>
> >It is important to emphasize that the primary goal of our research is to demonstrate the effectiveness of alignment through a two-player gaming framework, specifically designed to produce a safe LLM (defense model) that is robust to various attacks. The adversarial agent serves as a critical component during training; however, since both the adversarial and defense agents evolve over iterations, the framework also results in a strong attack model (adversarial agent). But we agree that  the adversarial agent could, in theory, be misused to generate harmful attacks.

---

> > ### Comment · Reviewer_6iyV · 2024-11-26
> >
> > Thank you for your response. My concerns about the performance w.r.t to the training amount and ethics statement have been addressed, but the first concern is not fully solved. Please see the discussion below.
> >
> > **Results to validate the convergence to NE**
> >
> > I agree that the Nash gap in Eq (3.6) can be used to evaluate convergence to NE, but I do not find any figure or table to validate that the gap is (approximately) zero. The authors mentioned Fig. 2 (c) in their rebuttal, but the gap between two iterations is not close to zero in Fig. 2 (c), which does not serve as evidence to validate the convergence to NE. Therefore, the current manuscript still lacks results to support the claim of convergence to NE.
> >
> > Moreover, I think the gap between the harmful rates of two iterations may not be a good approximation of the Nash gap for two reasons. First, the attack and defense agents are only trained for fixed steps (200 and 400), which is not the best response in Eq (3.6). Second, the diversity term should be removed from the objective of the attack agent when training for the best response. As the authors mentioned Fig. 2 (c) as an evidence for convergence, the value J only considers the attack success rate (or harmful rate) without the diversity. Therefore, the best response should trained to optimize only for the attack success rate. In conclusion, I think a better way to get the approximate best response to the learned defense agent is to fix the defense agent and train an attack agent *using the base model to maximize the attack success rate without diversity term until convergence*. The approximate best response of the defense agent can be learned similarly.

---

### Official Review · Reviewer_KfEA · 2024-11-04

**Soundness:** 2
**Presentation:** 3
**Contribution:** 3
**Rating:** 5
**Confidence:** 4

**Summary:**

- This paper proposes a 2 player adversarial zero-sum game (GPO) to develop a more robust and less toxic LLM. It consists of a defensive model that generates high quality and safe responses to the prompts generated by the adversarial agent. The adversarial agent generates prompts to try and make the defensive model generate bad or unsafe responses.
 - As a side effect, the adversarial agent serves as a good red-teaming partner.
 - A diversity parameter in adversarial agent’s reward ensures that a diverse set of prompts are covered during the training process
 - GPO hows strong improvements in safety alignment.

**Strengths:**

- The authors showcase the effectiveness of GPO and the diversity reward across different safety datasets and attacks.
- GPO does not seem to harm general quality despite only being used to align the model for safety scenarios.
- The paper is well written and method is clearly detailed.

**Weaknesses:**

- A pretrained Llama 2 7B model is used as a base, which then goes through SFT and RLHF. The data used for this isn't specified and it is unclear how the quality of the post-SFT model affect alignment. For example, [Vicuna 7B has a score of 6.00 on MT-Bench](https://lmsys.org/blog/2023-06-22-leaderboard/), which is comparable to the score post GPO.
 - The paper largely focuses on safety alignment and it is not clear how much GPO would benefit general alignment.
 - It is not clear how this method generalizes to larger models.

**Questions:**

- The typical RLHF objective anchors to the intial reference policy. It is not clear why the GPO objective anchors to the policy from the previous step and how this affects this.
 - Given that the anchor is updated at every step, this would result in a larger policy shift for both the defensive and adversarial agents. How does the RM perform when the prompts generated by the adversarial agent is OOD?

---

> ### Author Response · Authors · 2024-11-24
> **Official Comment by Authors**
>
> Thank you for your valuable comments! Based on your feedback, we have included the details about SFT data in Appendix B3. **The changes are marked in blue.**
>
> **Q1: A pretrained Llama 2 7B model is used as a base, which then goes through SFT and RLHF. The data used for this isn't specified and it is unclear how the quality of the post-SFT model affects alignment. For example, Vicuna 7B has a score of 6.00 on MT-Bench, which is comparable to the score post GPO.**
>
> > The SFT dataset used in our model follows the approach of Vicuna. It consists of 53k user-shared conversations across various domains such as mathematics, knowledge querying, and coding, which are collected from ShareGPT.com. This dataset size is slightly smaller than the 70k dataset used in Vicuna. The reason for having only 53k data is that the full 70k dataset is not accessible as it is not open source. Subsequently, some individuals have crawled and cleaned a 53k dataset. Despite being smaller than the Vicuna dataset, this dataset size still offers strong generalization capabilities. We have provided more details on the Appendix B3.
> We want to further emphasize that **the primary focus of our approach is safety alignment, with MT-Bench serving as a tool to verify that safety enhancements do not substantially compromise utility**. Despite relying on a potentially less powerful SFT dataset, our two-agent game framework improves the model's safety while ensuring that utility either remains stable or exhibits a smaller improvement when stronger SFT models are utilized.
>
> **Q2: The paper largely focuses on safety alignment and it is not clear how much GPO would benefit general alignment.**
>
> >Our primary objective in this work is to validate the feasibility of the two-agent game framework for alignment. Safety, in this context, is particularly well-suited to evaluation via the reward model (RM) because the RM can robustly judge whether a model's response adheres to safety norms. The adversarial and defensive agents present a strong game-theoretic scenario where both progressively improve through iterations, ultimately producing a robust defense model and a strong attacker model. This effectiveness is particularly evident in OOD datasets, as shown by the experiments in Table 1 and Table 2. We thank the reviewer for highlighting this interesting direction, and we plan to explore its effectiveness in general alignment further in the future.
>
> **Q3: It is not clear how this method generalizes to larger models.**
>
> >Our GPO framework is designed to be model-agnostic, as its reliance on policy optimization and diversity rewards makes it scalable. Although empirical validation on larger models is limited due to computational constraints, our theoretical analysis and the framework's reliance on established reinforcement learning principles indicate that it should scale well.
>
> **Q4: The typical RLHF objective anchors to the initial reference policy. It is not clear why the GPO objective anchors to the policy from the previous step and how this affects this.**
>
> >Our framework is an iterative process consisting of two agents trained alternately, whereas typical RLHF involves only one agent trained once. In our approach, each iteration corresponds to a complete RLHF process. Consequently, the initial reference policy should be the policy at the start of each iteration—which is the policy from the previous step.
>
> **Q5: Given that the anchor is updated at every step, this would result in a larger policy shift for both the defensive and adversarial agents. How does the RM perform when the prompts generated by the adversarial agent is OOD?**
>
> >Our RM is based on Llama-Guard [1], a toxicity classifier that evaluates model outputs based on their toxicity levels. The task it performs is relatively straightforward, as it only requires scoring based on the model's output. Llama-Guard has been specifically designed to classify toxicity across a broad range of inputs, and its performance has been demonstrated to generalize well to OOD prompts. As shown in the Llama-Guard paper, the model exhibits strong OOD generalization, meaning it can effectively handle and score toxicity in previously unseen or out-of-distribution data. Thus, we expect the RM to continue performing well even when the prompts generated by the adversarial agent are OOD.
>
> >[1] Hakan Inan, Kartikeya Upasani, Jianfeng Chi et al. Llama guard: Llm-based input-output safeguard for human-ai conversations. arXiv preprint arXiv:2312.06674, 2023.

---

> > ### Comment · Reviewer_KfEA · 2024-12-03
> > **Thank you to the authors**
> >
> > Thank you for your response! Many of my questions have been answered.
> >
> > > Our primary objective in this work is to validate the feasibility of the two-agent game framework for alignment. Safety, in this context, is particularly well-suited to evaluation via the reward model (RM) because the RM can robustly judge whether a model's response adheres to safety norms.
> >
> > I agree that safety alignment lends itself well to this setup and the safety results you have provided are encouraging. However it doesn't demonstrate that GPO extends well to alignment in general. To further validate the feasibility of the two-agent game framework for alignment more broadly, it would be valuable to see how this method generalizes across model sizes, especially since GPO is more computationally intensive than typical alignment methods.
> >
> > Further ablations on model sizes and a comparison with existing alignment methods would provide a more comprehensive understanding of the impact and potential of GPO.

---

### Meta-Review · Area_Chair_qTGD · 2024-12-20

**Metareview:**

This paper presents an interesting approach (GPO) for safety alignment in language models, demonstrating promising results on a range of safety benchmarks. However, we recommend rejection due to several key limitations. While the method is clearly presented and the experiments demonstrate the effectiveness of GPO for safety, the evaluation relies on a potentially weak base model (post-SFT Llama 2 7B) whose training data is unspecified. This makes it difficult to isolate the true contribution of GPO versus improvements inherited from the pre-training. Furthermore, the limited scope of evaluation, primarily focusing on safety alignment, leaves the impact on general alignment and scalability to larger models unexplored. These weaknesses, particularly the lack of clarity regarding the base model and limited scope, hinder the overall impact and generalizability of the presented work, warranting rejection for this venue.

**Additional Comments On Reviewer Discussion:**

Some concerns are addressed during rebuttal, but the paper will be benefitted with another iteration.

---

### Decision · Program_Chairs · 2025-01-22

Reject